# Pharmacy Staff Experiences and Needs During Second Dispense of Driving-Impairing Medicines: A Qualitative Study

**DOI:** 10.3390/pharmacy13050146

**Published:** 2025-10-09

**Authors:** Karin Benning, Liset van Dijk, Johan (Han) J. De Gier, Sander D. Borgsteede

**Affiliations:** 1Department of PharmacoTherapy, -Epidemiology & -Economics (PTEE), Groningen Research Institute, University of Groningen, A. Deusinglaan 1, 9713 AV Groningen, The Netherlands; karinbenning@live.nl (K.B.); l.vandijk@nivel.nl (L.v.D.); 2Nivel, Netherlands Institute for Health Services Research, Otterstraat 118, 3513 CR Utrecht, The Netherlands; 3Independent Researcher, 4904 ZR Oosterhout, The Netherlands; degiercs@planet.nl; 4Department of Clinical Decision Support, Health Base Foundation, Papiermolen 36, 3994 DK Houten, The Netherlands

**Keywords:** consultation, driving-impairing medication, patient communication, community pharmacy, patient information, driving safety, second dispense, qualitative research, pharmacy practice, pharmacist–patient interaction

## Abstract

Driving-impairing medicines (DIMs) are associated with an increased risk of traffic accidents. While Dutch pharmacy staff are expected to counsel patients at the first and second dispense of DIMs, current practice suggests that second-dispense consultations are underutilized. This study explored pharmacy staff’s experiences and perceived barriers in addressing driving impairment during the second dispense. Qualitative, semi-structured interviews were performed with 17 staff members in community pharmacies across the Netherlands. Transcripts were coded using thematic analysis in Atlas.ti, applying both deductive and inductive coding strategies to explore current practices and improvement needs. Participants reported that they provided detailed information on medication use, side effects, and driving impairment during the first dispense. In contrast, driving fitness was only discussed during the second dispense when patients initiated the topic, which rarely happened. Barriers to discuss DIMs included time constraints, a lack of protocols or prompts in pharmacy software, limited privacy, and patients’ reluctance to communicate about this topic. Many pharmacy technicians relied on closed questioning and observed a lack of patient initiative. Facilitators included strong patient relationships, access to medical records, and a desire for training in consultation skills. Pharmacy staff expressed the need for improved protocols, better ICT (Information and Communication Technology) integration, and targeted communication tools to support safe use of DIMs. In conclusion, second-dispense consultations for DIMs are underused and can support patients in safer medication use. Improved implementation will lead to better-informed choices about medicines and driving, and strengthen the pharmacy’s contribution to traffic safety.

## 1. Introduction

Previous research showed an increased risk of travel accidents associated with several commonly used driving-impairing medicines (DIMs), including benzodiazepines and opioids [1,2]. The accident risk estimated for benzodiazepines and Z-drugs (non-benzodiazepine hypnotics such as zolpidem and zopiclone) is two to three times higher for a serious injury and five to seven times higher for fatality; for opioids five to eight times higher for serious injury and five times higher for fatality [3]. Despite these substantial risks, patients might not recognize side effects such as dizziness, blurred vision, or reduced alertness and are often unaware of the increased accident risk [4].

Because of this increased risk, most countries prohibit driving under the influence of impairing substances, including medicines, but only a few refer to a list of banned substances. Penalties for drug-driving offences vary considerably between countries, ranging from license suspension of a few months to one year in others, and fines from hundreds to thousands of euros [5]. In the Netherlands, the setting of this study, citizens may not drive when using certain DIMs, and prescribers and pharmacists are responsible for informing patients [6,7].

Pharmacy staff need to inform patients when dispensing medication and distinguish counselling at first, second, and subsequent dispenses and inform patients about its risks [8]. Risk communication is important for effective decision-making in health-related behavior [9,10]. It helps patients to weigh the risks and benefits of the proposed treatment or behavioral change. Previous research shows that health care professionals do not always fully inform patients about the potential risks of driving-impairing medications [4,11,12]. The process of informing patients and discussing risks starts with the first dispense. This moment focuses on information about the use of the medicine, therapeutic effects, and potential adverse effects. The second dispense is an opportunity to discuss the patient’s first experiences with the medication [13]. Moreover, consultation at second and subsequent dispenses also covers issues such as changes in medication, reduced adherence, suspected pharmacotherapy-related problems, and any other questions regarding the patient’s medication use [14]. Pharmacists and pharmacy technicians both have the skills to conduct consultations, with pharmacy assistants generally having more experience and pharmacists having a higher level of education [13]. However, patient counselling during the first and, specifically, the second dispense can be improved, both in general [15,16,17] and specifically with regard to DIMs [12,18].

A model to use for patient consultation is the Calgary–Cambridge model for communication. In this model, communication skills are organized around five core tasks: initiating the consultation, gathering information, physical examination, explanation and planning, and ending the consultation [19]. Greenhill and colleagues studied the model in pharmacy practice and demonstrated that it can be applied to pharmaceutical consultations, in which the skill of physical examination does not play a significant role for pharmacists [20]. Pharmacy staff provide information at first dispense, but explore the patient’s expectations and experiences less frequently, leading to one-sided consultations in which the pharmacy technician dominates by providing information [17]. With respect to driving, information is usually given at first dispense. At the second dispense, pharmacy staff are expected to discuss the patient’s actual experiences with the medicine and identify any related problems [13,21].

Studies about the effect of patient counselling about DIMs are scarce, and the available information illustrates that patients are not well informed about the effects of medication on driving [22], and that the implementation of tools to support professionals needs more attention [18,23,24]. Good and available information can contribute to the awareness of the risk of driving under the influence of DIMs among professionals and patients [16,25]. The need for this is illustrated by a recent study, which suggested that older drivers involved in driving accidents did not use fewer DIMs after crashes than before [26]. This underscores the need to explore current consultation practices to improve patient behavior concerning DIMs.

In order to identify opportunities for the improvement of consultation, a good start will be to explore current practices with respect to patient information concerning DIMs. Community pharmacy staff can play a key role in patient consultation concerning medication use and driving safety. The moment of the second dispense is most suitable because the patient’s actual experiences can be discussed and used in shared decision-making concerning continuation of treatment.

The aim of this study is therefore to explore pharmacy staff’s experiences and perceived barriers in addressing driving impairment during the second dispense. This is to ultimately improve the effectiveness of these consultations, so that patients feel comfortable and empowered to express their experiences and concerns and make shared decisions with their health care providers.

## 2. Materials and Methods

### 2.1. Study Design and Setting

Qualitative, semi-structured interviews were conducted with pharmacists and pharmacy technicians in community pharmacies across the Netherlands. Pharmacy staff did not perform any intervention as part of this study. In the Netherlands, the pharmacy typically dispenses a two-week supply when a new medicinal treatment is started. Verbal instructions are provided at the time of dispense, supported by written information usually comprising (1) a drug label with usage instructions attached to each package, and (2) two types of written materials: the legally required manufacturer’s package leaflet and a patient-centered information leaflet printed in the pharmacy. If applicable, the pharmacy staff and patients decide on continuation of treatment two weeks after the start, at the first refill.

COREQ guidelines were used to report key aspects of the research team and reflexivity, study design data, and analysis and findings (see Appendix A) [27].

### 2.2. Participants

Pharmacy staff members with an interest in participating in research within the networks of the research institutions of the authors (LvD, Nivel, and SB, Health Base Foundation) were approached with the aim of selecting a diverse group of participants with respect to age, geographic location (urban/rural), and professional experience. A total of 75 pharmacies were contacted by email, of which 17 staff members were willing to participate. The remaining pharmacies either did not respond to the email or indicated they did not have time [28]. An appointment for the interview was arranged by email or by phone. Each pharmacy team was asked to select a pharmacist and/or a pharmacy technician involved in dispensing and consultation of patients. If the pharmacist or pharmacy technicians wanted to participate, an interview was scheduled.

### 2.3. Interviews

Semi-structured, in-depth interviews were conducted between February and May 2022. The interviews took place in a private setting (no one else was present), on-site in the pharmacy or by video call, depending on the preference of the participants, and the researcher made field notes during the interview. We aimed to perform 15–25 interviews until data saturation was reached.

Prior to each interview, the aim of the interview was explained, and the participants gave informed consent. At the start of the session, permission was obtained to audio-record the interview. All interviews were performed by the first author (KB), who was trained as an interviewer during the study, and was supervised throughout the process by two experienced qualitative researchers (LvD and SB). KB had no experience with qualitative research prior to this study.

Ethics approval was not required under the Dutch Medical Research Involving Human Subjects Act (Wet medisch-wetenschappelijk onderzoek met mensen, WMO), as the study involved only healthcare professionals, did not include patients, and did not affect any personal patient data [28,29]. Data were collected anonymously and stored in compliance with relevant privacy regulations.

### 2.4. Interview Guide

The interviews were guided by a topic list focused on pharmacy staff experiences and opinions regarding the provision of information about driving impairment at both the first dispense of DIMs and the information about driving at the second dispense.

Based on the literature and previous research by the study team, the first draft of the interview guide was prepared by the first author and reviewed by LvD and SB. During data collection, additional relevant topics emerged, and some questions were slightly rephrased. The interviews focused on two main areas: (1) the guidance at the first dispense of DIMs and (2) the need for information at the second dispense of DIMs. The guide also addressed oral and written communication, the structure of the consultation, and whether a dialogue occurred with the patient. In addition, the needs of the pharmacy staff regarding improvements to second-dispense practices were explored.

During the first five interviews, we assessed whether the topics in our guide were adequate to answer our research questions, and minor changes to the wording of the questions were made accordingly. All interviews were included in the analysis. The topic list is included in Table 1. The guide consisted of open-ended questions along with follow-up questions for further exploration.

### 2.5. Data Analysis

All interviews were audio-taped, transcribed verbatim, and made anonymous. To ensure textual correctness, the researcher (KB) read the transcripts while listening to the interviews. The transcripts of the interviews, including the field notes, then served as data. After 10 interviews, certain themes began to be repeated (data saturation). The next five interviews were already planned to confirm saturation, so that more nuanced details could be recorded and the data enriched. The researchers (KB, LvD, and SB) coded the first three interviews independently to identify key themes, using the themes from the topic list (deductive coding) and themes that the interviewees considered to be important as codes (inductive coding). At the end, a coding tree was made in Excel, combining deductive and inductive analytical strategies to enhance the structure and depth of analysis and to contribute to the robustness of the findings [30]. Coding itself was performed in Atlas.ti 22. In the subsequent transcripts, these codes were further explored until additional interviews provided no new information with respect to the research question. Citations covering the topics were included in the analysis. One of the researchers (LD) translated the quotations from Dutch to English; the other authors checked the translation.

To identify cues for improving the effectiveness of consultations about driving-impairing medication, no formal theoretical framework was defined prior to the study. During the analysis, we applied the COM-B model to the findings. The analyses were performed by KB (pharmacy student, female, BSc) under the supervision of two researchers with experience in qualitative methods: LvD (experienced researcher in pharmacy practice research with a background in social sciences, female, PhD) and SB (experienced researcher in the development and implementation of pharmaceutical interventions with a background as pharmacist/epidemiologist, male, PharmD/PhD). HG (independent researcher, male, PharmD/PhD) provided methodological and interpretive comments, drawing on his extensive experience in research on driving-impairing medication. All researchers had a scientific interest in the topic without specific assumptions about the content. During the analysis, the authors ensured the validity of the results by critical discussion and looking for cases that seemed to verify or conflict with the findings from the interim analysis. Statistical methods were not used to calculate levels of agreement.

## 3. Results

In total, 17 pharmacy staff members were interviewed in 15 interviews, and in two interviews, two pharmacy staff members participated. Table 2 shows the characteristics of the participants. Seven pharmacists and ten pharmacy technicians participated, their ages ranging from 24 to 57 years. On average, they had 15.5 years of experience in the pharmaceutical practice, with pharmacy technicians having longer working experience than pharmacists. Fifteen participants were female, and two were male. Eight pharmacies were located in rural areas and seven in urban areas. Interviews lasted approximately 30 min, ranging from 25 to 40 min. In total, eight interviews were performed on-site and seven interviews by video calls.

We will present our findings according to the three main themes of our topic guide: (1) the provision of information, (2) barriers to providing information at second dispense, and (3) facilitators for providing information at second dispense. In cases where experiences differed between pharmacists and pharmacy technicians, or between urban and rural areas, these are reported. No differences were observed based on gender or years of work experience. The quotations included illustrated typical views expressed by pharmacy staff. The analysis did not reveal that the results were influenced by the characteristics of the researchers. The coding tree is presented in Appendix A, with most codes being the result of deductive coding.

### 3.1. Provision of Information on Driving-Impairing Medicines

#### 3.1.1. First Dispense

The interviews showed that pharmacy staff mainly provided practical information to the patient at the first dispense of DIMs. This practical information contained an explanation of how the medicine worked, the user instructions for the medicine, any side effects that might occur, and advice about driving. Driving-related advice addressed whether patients could drive, the duration of impairment, and when driving could resume.


*At first dispense you always tell the usual things the effects of the medication, how people should take it, and, well, with certain doses what they should be careful about when they shouldn’t drive, and when they might be able to drive again. (Pharmacy technician, female, 13 years of experience)*


Pharmacy staff did not use the formal descriptions of the classifications for driving risks but instead translated the information into patient-friendly language. Six participants mentioned that driving under the influence of impairing medication is just as illegal as drinking alcohol and driving.

As part of counselling, patients received relevant written and online information at the first dispense. This information included the package leaflet, a video with information about the medicine (*WatchyourMeds^®^; Dutch: Kijksluiter^®^*) [17], and/or an individualized information leaflet printed on demand. Some patients also received the DIMs delivered to their homes, or patients collected their medicines via pharmacy lockers. In that case, pharmacy technicians called the patient to provide verbal information that would otherwise be provided at the counter in the pharmacy. These patients also received the same written materials.


*The standard VI folder is given along with a Watchyourmeds^®^-code [Standard for video instructions; author explanation] they can scan to see advice, and I assume that driving skills recommendations are also included. But, otherwise we do not provide a special booklet with medicines that pose a risk to driving. (Pharmacist, female, 7 months of experience)*


Two pharmacy technicians stated that they observed that telephone consultations were more effective than those at the counter. They experienced that there was more interaction due to increased privacy and less pressure.

#### 3.1.2. Second Dispense

At the second dispense of DIMs, the pharmacy staff stated that they did not discuss driving impairment unless patients raised the issue themselves. Questions asked of patients were usually general in nature. Examples of questions asked were “how are you doing”, and “how did you experience the medication and did any side effects occur?”. Also, patients did not always want a consultation at the pharmacy counter. Unlike the first dispense, patients who had their medication delivered or who collected it via the locker were not called for a second-dispense consultation, except in one pharmacy.

The interviews showed that pharmacy staff paid more attention to the first dispense than to the second dispense of DIMs. Support or information provided to the patient at the second dispense did not always occur, for example, due to rush hours in the pharmacy. Four respondents indicated that there was minimal or no attention to the second dispense at all. Furthermore, no written information was provided at the second dispense, except the package leaflet in a full package. This applied to all types of DIMs. Nine respondents used instructions with support for consultation at the first dispense and with general instructions at the second dispense. The pharmacy staff experienced that this could sometimes help to give the patient all the information they need.

*No, I don’t think driving skills are actually discussed at second dispense, unless the patient asks, well, that is possible of course, then they say: “*I still experience some side effects, drowsiness or dizziness. Can I drive?*” Of course you encounter people who ask such questions, however I don’t think it’s a topic we typically bring up ourselves. (Pharmacist, male, 6 years of experience)*

#### 3.1.3. Ideal Second-Dispense Consultation

Pharmacy technicians and pharmacists consider, ideally, a second-dispense consultation more patient-centered and include patients’ experiences concerning side effects. Also, in an ideal situation, they would ask the patients to talk about his/her experiences with the medicines. When patients leave the pharmacy, they should leave the pharmacy well-informed and reassured. Pharmacy staff emphasized the importance of convincing patients that DIMs can make driving dangerous and that medication may affect daily life. Although pharmacy staff were motivated to discuss DIMs, they found it difficult to convey this information, as they thought patients might feel controlled.

Also, pharmacy staff would like to have an open and trusted consultation at the second dispense of DIMs, so the patients can ask questions and express their concerns. However, this rarely occurs in practice. Pharmacy staff did not know why these consultations did not occur. It might be due to consultation style, where closed questions dominate, but also because of insufficient trust between pharmacy staff and patients.


*An ideal consultation would be where you can talk openly [about driving impairing medication], and that the patient is of course open to it as well. That is important. But as a pharmacy technician, you can of course try to create that openness by asking the right questions. (Pharmacy technician, female, 36 years of experience)*


### 3.2. Barriers to Providing Information at Second Dispense

#### 3.2.1. ICT

The major pharmacy IT system (Pharmacom^®^) provided brief on-screen instructions with support for first- and second-dispense consultations for pharmacy staff. While these instructions for the first dispense were generic, the instructions for the second dispense contained only general questions, lacking prompts related to driving fitness. This might be a reason why less attention was paid to driving during the second-dispense consultation.

#### 3.2.2. Workload

When it was busy in the pharmacy, less attention was paid to the second-dispense consultations of all medicines, including DIMs. Patient impatience due to long waiting time and time pressure experienced by pharmacy technicians led to short consultations and had a negative impact on the quality of these consultations. The lack of attention to the second-dispense consultation was also influenced by pragmatic financial considerations. Unlike first-dispense consultations, pharmacists do not receive extra reimbursement for second-dispense consultations.

#### 3.2.3. Privacy

Some pharmacy staff struggled to ask patients personal questions, for example, about their work or private situation. Not all patients seemed to be open to answering questions about their private lives. Yet, this information was sometimes necessary to provide tailored information. Pharmacy staff found it difficult to assess which questions patients considered to be private and which questions not. For this aspect, we found a difference between pharmacies located in rural and urban areas. Pharmacy staff in rural areas experienced that patients were more open to a consultation at the counter than patients who live in urban areas. They felt that they had an advantage due to a closer relationship with their patients. One of the participating pharmacies was owned by a dispensing general practitioner (GP). Here, the pharmacy staff did find it an advantage that they knew the patients well, and the GP could easily be consulted.


*We often have some background information about our patient, so that that does allow you to enter a bit more into what you are working on, say what you are asking about. I think this is also because we are a bit more village-like here so we know our population quite well and I think that makes it, we do have a real bond of trust with a lot of patients. (Pharmacy technician, female, 13 years of experience)*


In addition, pharmacy staff in rural areas have had access to medical records. Moreover, the rural setting resulted in more personal contact, illustrated by the quotation below:


*Because it’s such a small village, people come to the counter with questions or they call, so communication lines are very short. A big advantage of being a dispensing GP is that you have short lines with the GP and you have access to the medical file. You also know the patients very well, because we don’t have that many patients. You see people very often and at a certain point you know their names, so that’s an advantage. (Pharmacy technician, female, 11 years of experience)*


Lack of privacy is experienced as an obstacle in all pharmacies, yet pharmacy staff also mentioned that there was a consultation room in all pharmacies where discussions could be held in private. This facilitated that patients who preferred to discuss their medication in private could do so, yet this room was underused. Some pharmacies had additional privacy features such as separate collection desks or soundproof counters. As a result, the consultation between the patient and the pharmacy staff could not be heard by the other people in the waiting room, which ensured more privacy.


*That’s the tricky part of course, you get an audience that easily finds you: you’re the most accessible healthcare provider there is, you can just walk in here. At a general practice, you need to make an appointment these days you can’t get in otherwise. But at the same time, we should not be too open, because it involves privacy matters of course. (Pharmacist, female, 13 years of experience)*


Another barrier for a dialogue at the counter was that the fact that the patient and the pharmacy staff member did not take or have the time for a consultation. If patients are not open to a consultation, the pharmacy staff can observe this not only from the body language of the patient but also by the brief answers, as if the patient is already “outside the pharmacy”, mentally. Also, pharmacy technicians experienced that patients believed that they already understood the medicine, especially if counselled by their GP. In addition, some patients felt ashamed of their medication use and therefore wanted to leave the pharmacy as soon as possible. Often, the pharmacy technician then thought that there was no point in explanation: The medicines were given to the patient, and the second-dispense consultation was practically omitted. After providing basic information, people had been made aware that their reaction may be impaired. After that, they considered it the patient’s own responsibility.


*Then, for example with sumatriptan, they know that it is only safe to go and drive four hours after intake, but some still take it and drive home. They know the risks, but choose not to act on them. And in order to get through that and say it’s really not safe to drive, I notice that the pharmacy technicians find that difficult. (Pharmacist, female, 7 months of experience)*


### 3.3. Facilitators for Providing Information at Second Dispense

#### 3.3.1. Verbal Support and Education

Four respondents indicated that they see added value in continuing education to improve knowledge about DIMs and consultation skills at the second dispense. Refresher training in consultation skills can create more awareness among the pharmacy staff, for example, to ask open questions and to discuss driving impairment at the second dispense. Refresher training could also include how to discuss sensitive topics, ask personal questions, and how to engage resistant patients who are not open to consultation. So that the pharmacy staff can be taught how to deal with these situations.

Four participants emphasized the value of refresher training in DIMs and consultation skills, especially for discussing sensitive topics and engaging resistant patients. WatchyourMeds^®^ was noted as particularly helpful for low-literacy and non-Dutch-speaking patients.

One pharmacy technician also saw added value in videos with information about a specific medicine for patients. They can see the information again at home, and this is especially helpful for limited literacy and non-Dutch-speaking patients.


*We are increasingly dealing with people who don’t speak the language well. In that case, Watchyourmeds^®^ is ideal, because they can simply look at an animation video in their native language. That way, you know that the information will come across clearly. (Pharmacy technician, female, 25 years of experience)*


One pharmacy technician indicated that it would be facilitating if she knew the indication for medication use so she could tailor the information at the first and second dispense of the medicine. According to her, the indication of the medication makes a difference, and if the indication is known, the consultation can be adjusted accordingly. In addition, it must also be clear to the pharmacy technician whether it concerns a second dispense, so that the pharmacy technician at the counter can respond to this.

Other facilitators that were mentioned included more attention to second dispense during the basic training of pharmacy technicians, increasing public awareness of the pharmacy’s role and accessibility, and implementing follow-up consultations with patients three to six months after initiating treatment.


*Ideally, I’d like to ask patients again after three months how they are doing with their medication. We currently don’t have anything for that, and I think that it’s almost even more important than the second dispense. (Pharmacy technician, female, 30 years of experience)*


#### 3.3.2. Support for Pharmacy Staff

Ten respondents recommended that they would like instructions with support for consultation for the second dispense that includes specific prompts about driving fitness. Two respondents suggested adding visual aids to prompt conversation. Three respondents suggested a flowchart or decision tree for patients to consult when unsure about the effects of medication on driving. This can ensure that patients contact the pharmacy sooner in case of problems and find a solution more quickly.


*You would prefer just a A6 with a short explanation about driving fitness… most people don’t realize how these medications affect reaction time… You need to make the connection to things like driving, mowing the lawn, or even that one little step before going up the stairs. It is not just in traffic. (Pharmacy technician, female, 16 years of experience)*


## 4. Discussion

Pharmacy staff provide detailed information on the effects of medication on side effects and driving impairment during the first dispense. In contrast, they seldom discuss driving fitness during the second dispense unless patients initiate the topic.

Barriers to adequately addressing driving fitness during the second dispense were primarily related to the opportunity component of the COM-B model [31]. Contextual factors such as time constraints and limited privacy at the counter restricted opportunities for meaningful discussion. From the pharmacy staff’s perspective, patients’ reluctance to engage in conversations about this topic further compounded the challenge. In addition, the absence of structured protocols left staff without evidence- or practice-based guidance to rely on.

A capability-related barrier concerned communication skills: Pharmacy staff often depended on closed questioning, which limited open dialogue. At the same time, several facilitators were identified. These included strong patient–pharmacist relationships and a clear willingness among staff to refresh their consultation skills—both of which support discussions on sensitive topics such as driving fitness. Pharmacy staff indicated that they recognized the added value of second-dispense consultations, and aspects related to motivation did not emerge as a barrier. Figure 1 presents the findings of this study from the perspective of the COM-B model. To address these challenges, pharmacy staff emphasized the need for improved protocols, better ICT integration, and targeted communication tools to support the safe use of DIMs.

No different opinions between pharmacists and pharmacy technicians were found. Pharmacy technicians had more experience with (second dispense) consultations and could provide more examples from these experiences. Pharmacists based their opinions more frequently on observations of consultations in their pharmacy and paid more attention to organizational aspects. This is in line with their professional roles, where pharmacy technicians have a more executive role and pharmacists have final responsibility and more pharmacological knowledge [17,32]. In case second-dispense consultations are stimulated, pharmacy technicians will be involved more frequently compared to pharmacists, as they have a greater executive role. On the other hand, pharmacists may expect more complex questions about DIMs, for example, about drug–drug interactions or alternatives for DIMs.

We found that in rural settings, patients were more open to a consultation about driving-impairing medication. Pharmacy staff felt that communication was improved due to the closer relationship with their patients. Some other studies also confirmed that in more rural areas, physicians know patients on a more personal level and hence are more engaged in socioemotional communications [33]. This more personal approach seems important when advising on DIM. Decisions about whether or not to drive a car are very much related to personal lifestyle choices. General views on (traffic) safety and practical considerations for organizing alternatives play a role. A good discussion on this subject must therefore allow room for both pharmacological considerations of the medicine and the patient’s personal situation.

Previous research has demonstrated that effective patient information about DIMs has the potential to reduce traffic accidents [4]. Our study shows that in the Netherlands, pharmacy staff inform patients properly during the first-dispense consultations, but counselling is usually absent during the second dispense. In a narrative review, Illardo and Speciale (2020) argued that while the use of effective communication skills is essential for adequate assistance and advice to patients, they need to strengthen the loyalty-based relationship with their patients [16]. This finding is also stressed by Te Paske et al. (2023), who argued that pharmacy staff should strive to enhance their patient-centered communication skills to build and maintain trust with patients [34]. While limited attention is seen for the second dispense of medication in general [35] for DIMs, this counselling is even more relevant due to reduced driving fitness [17]. Furthermore, pharmacy staff often do not encourage exchange of the patient’s perspectives on and experiences with medication [17], and if pharmacists engage in exploring medication use, it appears to occur on an ad hoc, rather than a systematic basis [36]. Our findings reflect this ad hoc exploration of DIMs.

One important question to raise is what responsibility patients can take to be well-informed about their medication. In general, patients perceive it as the pharmacist’s responsibility to guide patients and provide information, a view that is shared by pharmacists themselves [26]. Pharmacy staff in our study also emphasized the importance of addressing driving impairment during second-dispense consultations. For instance, if driving was contraindicated during the initial treatment, the second dispense provides an opportunity to assess whether it is now safe to resume driving according to them. However, suggesting that pharmacy teams should simply implement second-dispense consultations underestimates the complexity involved. A significant organizational barrier also found in previous research is the need for sufficient time and privacy to enable meaningful exchange of information and patient experiences [37,38]. In current practice, such conditions are challenging due to staffing shortages, logistic challenges, and the increasing commercialization of pharmacies, with less space for private patient consultation [39]. Also, in order to share experiences on medication use, there needs to be a trusting relationship between pharmacy staff and patients. Previous research showed that trust of patients in the pharmacy team was moderately high, indicating room for improvement, for example, by consistently applying patient-centered communication [34,40]. Best practices also show that elements of patient-centered communication can be successfully implemented, such as strengthening the relationship, gathering and providing information, shared decision-making, and enabling treatment success [41,42].

### 4.1. Strengths and Weaknesses of the Study

A key strength of this study is the inclusion of both pharmacists and pharmacy technicians as participants. In the Netherlands, pharmacy technicians work more often at the counter than pharmacists. Pharmacy technicians more frequently engage in patient encounters compared to pharmacists, yet their perspectives offer other insights into second-dispense consultations for medications that may impair driving ability. In addition, pharmacists and pharmacy technicians from pharmacies in rural and urban areas participated in the interviews. Distances to facilities are often longer in rural than in urban areas, and often with limited public transportation. This means that patients from rural areas are more dependent on using the car than people living in more densely populated areas is something important in relation to DIMs. Also, closer social ties within rural areas may facilitate more personalized consultations at the pharmacy counter. A final strength was that data saturation was reached with the interviews conducted, supporting the robustness of the qualitative findings.

This study also had some limitations. The first limitation of this study is the uneven distribution of work experience between pharmacists and pharmacy technicians. On average, pharmacy technicians in our study had a long working experience, while, on the other hand, pharmacists had, on average, a shorter working experience. Although data saturation seemed to be reached, it is possible that within both groups there were more opinions than we captured. Consequently, it is possible that the differences that were found between pharmacists and pharmacy technicians might be explained by the differences in working experience.

In addition, selection bias is possible, as participation was influenced by the respondent’s motivation to participate in research. Also, we did not check if participants had a specific interest or knowledge in the topic being studied. However, we did not notice specific interest in the topic being studied and do not expect that the opinions concerning driving-impairing medications were related to their interest in participating in research. Other limitations are related to the interview process. We did not ask participants to validate or comment on the interview transcripts. At the end of the interviews, respondents indicated that there were no additional topics requiring attention. Therefore, we believe that the research topic was explored in sufficient depth. Also, the use of video-call interviews limited our ability to observe the physical and social environment of the pharmacy setting and to establish rapport and trust with interviewees [43]. Other research showed that the data quality of video interviews is generally comparable, yet more sensitive topics are better suited for face-to-face interviews [44]. In our research, we did not find that the dispensing of DIMs was a sensitive subject, and we had similar conversations in the video and face-to-face interviews.

Despite these limitations, we believe that they do not compromise the meaning of the findings from this study.

### 4.2. Significance for Daily Practice

This study shows that pharmacy staff do not easily enter into the second-dispense consultation for medicines that may impair driving availability. As a consequence, patient guidance is not optimal for DIMs. One reason for the omission may be the lack of implementation of supporting tools for second-dispense consultation compared to protocols for first dispense In the Netherlands, for the second dispense, a general question set is available that supports shared decision-making based on the experiences of the patient during the first two weeks [35]. This so-called TRIAGE tool might be an interesting starting point to develop a specific protocol for consultation for DIMs at the second dispense. Together with improved communication skills such as listening effectively, eliciting the patient’s perspective, and bringing structure to consultations, this may empower pharmacy staff to conduct more effective second-dispense consultations [20,45]. Finally, integration of consultation prompts within pharmacy information systems could provide additional technical support, reinforcing patient-centered communication. A strategy that merges second-dispense consultation prompts with communication skills training could strengthen the safe use of DIMs.

Another promising approach is to empower the patient and reduce their dependence on health care professionals for decision-making concerning DIMs. Achieving this requires further improvement of patient-centered information concerning DIMs, not only with respect to usability, but also addressing health literacy and providing personalized content tailored to individual patient characteristics [46]. Opinions about medication and driving are also dependent on culture; hence, it is important to take the cultural background of the patient into account to tailor [47,48]. If possible, comprehensible information is presented in the native language of the patient, in the preferred format according to the patient.

The combination of driving and using impairing medication requires careful balancing between traffic safety risks and the benefits of enabling patients to continue driving, considered from both individual and societal perspectives. From the patient’s viewpoint, the ability to drive improves quality of life by remaining active in their personal and professional activities [49]. From a societal perspective, individuals who are able to drive remain active in the workforce and contribute to overall productivity.

### 4.3. Significance for Research and Policy

Currently, a second-dispense consultation is not routine practice. It is valuable to assess whether implementing such a consultation, with specific aims defined for first- and second-dispense counselling, would improve patient support. First-dispense consultation should aim to establish trust and rapport, provide pragmatic guidance to the patient, and engage in shared decisions concerning the process. Patients and pharmacy staff agree on the patient’s tasks for evaluating medicine use and perceived benefits/adverse effects, and on the staff’s role in using the patient’s evaluation as the starting point for the second-dispense consultation. During the second-dispense consultation, the treatment should be reviewed, the patient’s experiences should be discussed, and shared decisions should be made regarding modification, continuation, and monitoring.

Because this approach has not yet been implemented, future research should first explore patients’ perspectives. It is also important to determine how patients can be informed without increasing pressure on workforce capacity, as staffing levels are an important and growing constraint [39]. An interesting development in this context is the use of decision aids, which have shown promise in facilitating communication and supporting informed decision-making in the area of medication and driving safety [50]. Providing patients with relevant information at home would prepare them for the consultations in the pharmacy and hence support both patients and pharmacy staff in shared decision-making. When developing support materials, language and health-literacy specific demands must be considered and studied—for example, video animations and pictograms. Face-to-face consultations should be prioritized for the most complex patients, whether due to pharmacotherapy or other needs for personal assistance. This shift will require enhanced communication skills to be incorporated into training programs.

## 5. Conclusions

Second-dispense consultations for DIMs are an underused aspect where pharmacy staff can support patients to translate information into safer medication use. Moving toward structured, shared decision-making at the second dispense can help patients appraise their early experiences, weigh benefits and risks for driving, and agree on concrete next steps. This shift—delivered with health-literacy-sensitive communication and tools—has clear public-health relevance: Better-informed choices about medicines and driving can reduce drug-impaired driving while preserving mobility for patients who depend on driving. These changes can strengthen the pharmacy’s contribution to traffic safety, improve patient autonomy, and inform policy that supports a patient-centered pharmacy practice.

## Figures and Tables

**Figure 1 pharmacy-13-00146-f001:**
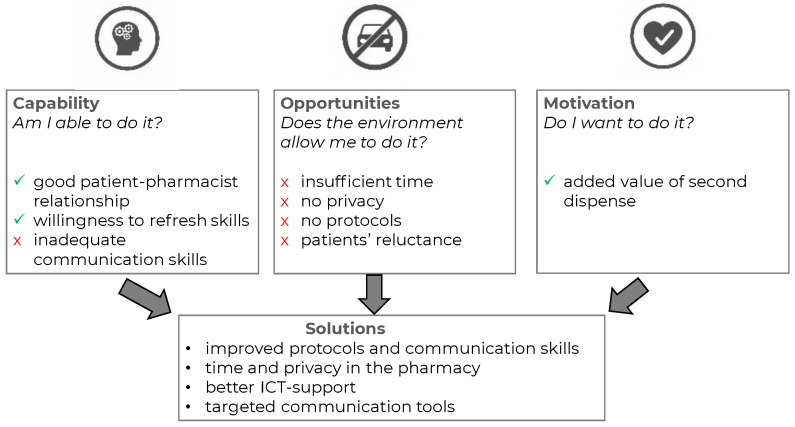
Main findings presented in the COM-B model with potential directions for solutions.

**Table 1 pharmacy-13-00146-t001:** Topics of the interview.

Main Theme	Example Questions
Guidance at the dispense of DIMs, first and second dispense	When a patient comes to collect a DIM, what is the guidance from your pharmacy?
Barriers providing information at the second dispense of DIMs	What factors hinder you when having a second-dispense conversation with patients about DIMs? Can you explain why?
Information needs at the second dispense of DIMs	What are your needs during the second-dispense conversation about DIMs?
Ideal second-dispense consultation	What would be your ideal second-dispense consultation for DIMs?
What do patients prefer or do not like at the consultation	What do you think patients appreciate during a second-dispense conversation about DIMs? Or what do they not appreciate?

**Table 2 pharmacy-13-00146-t002:** Characteristics of the participants.

Profession	
Pharmacist (n)	7
Pharmacy technician (n)	10
Gender	
Female (n)	15
Male (n)	2
Years of experience	Mean: 15.5 years (7 months–36 years)
<5 (n)	3
>5 (n)	14
Age	Mean: 40 years (24–57)
Geographical area	
Urban (n)	8
Rural (n)	9

## Data Availability

The data presented in this study are available on request from the corresponding author due to the nature of the data that have been collected. All interviews were anonymized, yet they still contain personal details, making the data not suitable for open availability.

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
