# Peer review of "Pharmacy Staff Experiences and Needs During Second Dispense of Driving-Impairing Medicines: A Qualitative Study"

_pharmacy, 2025, doi:10.3390/pharmacy13050146_

Round 1
Reviewer 1 Report
Comments and Suggestions for Authors
Comments are in the attached file.

Reviewer 2 Report
Comments and Suggestions for Authors
Peer Review: Manuscript pharmacy-3771731
The study addresses a timely and underexplored topic in the field of pharmacy practice.
Pharmacy staff in the Netherlands usually explain how certain medications can affect driving skills when they first give them to patients. However, they rarely discuss this issue during the second visit (unless the patient requests it). The main reasons identified are lack of time, lack of reminders in their IT systems, lack of confidentiality and the fact that patients do not want to talk about this topic. Staff also stated that they need better tools, clear steps to follow and more training to help patients discuss this topic more easily and in full confidentiality.
The following comments are intended to support the authors in strengthening the manuscript's clarity, methodological rigour, and overall impact.
General Comments
1. Consider grounding their qualitative study in a relevant theoretical framework from health communication or behaviour change (e.g., the Theory of Planned Behaviour or the Health Belief Model).
2. The study includes only 17 participants (7 pharmacists and 10 pharmacy technicians), with an uneven distribution of professional experience—technicians had more years of practice on average; the small number of participants limits the generalizability of the results and introduces selection bias.
3. Conduct a more systematic comparison between rural and urban settings, as well as between pharmacists and pharmacy technicians, to enhance the depth of the findings and provide greater insight into how contextual and professional factors influence second dispense practices.
4. Elaborate on the methodological implications of conducting video interviews (discussing how the lack of access to non-verbal cues or the physical context may have influenced data richness, and interpretation would strengthen the transparency and credibility of the study).
5. Explore how language or cultural differences affect communication about DIMs (the issue of non-native speakers is briefly mentioned).
6. Expand discussion on the practical implementation of the proposed improvements (e.g., protocols, training, ICT integration). Outlining potential strategies, required resources, and feasibility considerations would enhance the applicability and impact of their recommendations for pharmacy practice.
7. The manuscript lacks figures or diagrams, which could enhance clarity, especially a conceptual model or summary figure of barriers/facilitators (example: study workflow).
8. No supplementary materials are mentioned; if the interview guide or coding tree exists, it could be added as supplementary content.
Introduction
9. Some points are repeated in the manuscript (e.g., the importance of second dispensing, the role of pharmacy staff).
10. The text could be more concise and better structured into thematic paragraphs.
11. The introduction should benefit from referencing a communication or behavioural theory (e.g., patient-centred care, shared decision-making) to frame the problem.
12. A brief comparison with international practices is missing. Only 13 references are cited in the introduction, which do not sufficiently reflect the context of the study.
13. The transition from background to research aim is quite abrupt; a sentence summarising the knowledge gap and justifying the qualitative approach would help.
Materials and Methods
14. The manuscript should include or describe the coding tree structure (e.g., main categories, subcategories, example codes). It is necessary to create a table or figure in the manuscript or as supplementary materials.
15. The manuscript does not report any measure of agreement (e.g., Cohen's kappa) or how discrepancies were resolved.
16. There is no indication that participants were asked to review or validate the interpretations, which could have added to the trustworthiness of the findings. If there is a limitation of the study, it should be mentioned.
17. The authors do not reflect on their own potential biases or positionality, which is increasingly expected in qualitative research. This aspect should be completed/discussed in the manuscript.
18. Although the coding was both deductive and inductive, it is unclear whether any theoretical model (e.g., patient-centred communication, behaviour change theory) guided the deductive codes. This aspect should be commented on in the manuscript.
Results
The results are briefly presented, considering that only 15 interviews were conducted. Here, a situation of the codes identified in each interview would have been necessary.
Discussions
19. Use the Calgary-Cambridge model or the COM-B model (Capability, Opportunity, Motivation – Behaviour) to interpret communication challenges.
20. Discuss how pharmacy staff can support patients in becoming active participants in their care.
21. Briefly mention how second dispensing practices differ in other European countries or healthcare systems. The limited number of references used in this chapter denotes poor research on the topic in the specialized literature.
22. Suggest studies that could explore patient perspectives, test interventions (e.g., training modules), or evaluate ICT tools.
23. End with a clear, impactful statement about the need for systemic change, not just individual-level training. A guide for pharmaceutical staff could even be outlined that would concern second dispense of DMSs.
Conclusions
24. The conclusion restates points already made in the discussion without adding synthesis or emphasis.
25. Phrases like "there is room for more improvement" are vague and should be replaced.
26. The conclusion should highlight the potential impact on public health, pharmacy practice, or policy.
27. Suggest specific areas for further investigation (e.g., patient perspectives, intervention studies, international comparisons).
28. The conclusion should be strengthened by tying back to concepts like shared decision-making, health literacy, or behavioural change.
References
Generally speaking, the studied and cited references in this study are insufficient.
29. Add international comparative studies on DIM counselling practices.
30. Cite communication and behavioural science frameworks.
31. Include recent reviews on pharmacy-based interventions for medication safety or driving risk.
32. Reference qualitative research standards.
33. Incorporate patient-centred care literature to support the discussion on shared decision-making and trust.
34. While references are present, they may not fully follow MDPI's citation style (e.g., missing DOIs, inconsistent formatting).
Minor Issues and Formatting
35. Table 1. The table is presented as a single-column list. A two-column format (e.g., "Main Topic" and "Example Questions" or "Rationale") could enhance readability and informativeness (expand the table to include example interview questions or coding categories derived from each topic; indicate whether the topics were predefined (deductive) or emerged during interviews (inductive); consider linking each topic to the research objectives or themes in the results section for better coherence).
36. Line 83. Three points were forgotten at the end of the paragraph.
37. The manuscript appears to follow a general structure but may not strictly adhere to the MDPI formatting template (e.g., layout, font, spacing, figure placement).
Author Response
Please see the attachment uploaded for reviewer #2.

Reviewer 3 Report
Comments and Suggestions for Authors
This is a very interesting and well-written manuscript about an underexplored topic. the findings are relevant to pharmacy practice and patient safety. However, several modifications are needed to improve clarity, consistency, and methodology
Line 83: The use of "..." at the end of the sentence appears to be either a typo error or a missing reference. Please clarify.
Line 106: The abbreviation “PT” (pharmacy technician) is introduced without prior definition. Ensure it is spelled out at first mention.
The manuscript states that saturation was reached at the 10th interview, but this is mentioned much later in the methods section. It should be described earlier when introducing sample size justification. Also, it was not clarified why the author kept interviewing until 15 if saturation reached at 10.
The recruitment strategy is insufficiently described. Why were these 15 pharmacies chosen? why these pharmacy were approached? Was the sampling purposive, convenient, or random? much more information needed about the sampling. Provide rationale for site selection and recruitment.
Interview guide, it is good that it was dynamic and flexible and improved while conducting the study.
Table 2:
No need to mention gender twice! They are repeated, also stay consistent (male, female) or (Men, women), not (Women, Male)!
"Years of experience": clarify that "15.5 years" is the mean
Geographical area, here you used N = ##, but other rows were without N
there is an inconsistency in the way of writing
Interviews were quoted in English. Were interviews conducted in Dutch or English? If Dutch, describe the translation process and whether validation or back-translation was performed to preserve meaning.
Clarify how confidentiality and anonymization of participants’ data were ensured during analysis and reporting.
ICT (Line 241) is introduced without explanation. Spell out “Information and Communication Technology” at first mention.
Figures are absent. Consider including a thematic diagram or conceptual framework summarizing the main barriers and facilitators for second-dispense counseling to enhance reader understanding.
The discussion references international studies, such as reference 16 (Jordan), but integration of these findings is vague and without a clear context. Provide clearer context and comparisons with international data to strengthen the relevance of findings beyond the Netherlands.
All the previous comments qualify for Minor Revision. However, there is a critical issue in the methods and findings that made the recommendation (Major Revisions); the Validation.
Although the manuscript mentions "During the analysis, the authors ensured the validity of the results by critical discussion and looking for cases that seemed to verify or to conflict with the findings from the interim analysis." but this is insufficient
Validation and Reliability are essential in qualitative research. How were the interviews validated? (triangulation, member checking, respondent validation, inter-coder agreement...)
There is no mention of how reliability was ensured, especially since there are different ways of interviewing.
(in addition to the translation validation mentioned earlier)
Author Response
Please see the attachment uploaded for reviewer #3.

Reviewer 4 Report
Comments and Suggestions for Authors
This manuscript addresses a notable gap in pharmacy practice — the lack of structured consultation during second dispenses of driving-impairing medicines (DIMs). It is important for improving pharmaceutical care and patient safety. However, the article could benefit from a clearer presentation of results, a more thorough justification of claims, and increased methodological transparency. Here are suggestions to enhance your manuscript:
- Lines 55-58: Is there a protocol for how you provide consultation? And evidence that you have consulted patients?
- Line 64: explain the sentence – “This can lead to one-sided consultations.” Authors should specify whether this refers to a lack of shared decision-making, failure to elicit patient feedback, or a pharmacist-dominated monologue.
- Line 67: “k” delete
- Line 83: delete “…”
- Check consistency in terminology: some sections use “PTs” and others use “pharmacy technicians.” Maintain uniformity.
- Line 114-116: Did you exclude any questions from the interview? Include a brief explanation in the Methods section about any modifications to the topic guide during piloting and the reasons for exclusions or rephrasing.
- Line 155: Can a pharmacist and a pharmacy technician provide the same level of counselling? Discuss role differentiation in the Discussion or Methods. Regulatory responsibilities, educational background, and consultation authority should be compared.
- Line 255: Are consultations provided in a separate room where privacy is guaranteed?
- Mayor comments: Generally, the Results section is not presented clearly and does not clearly distinguish between pharmacists' and pharmacy technicians' consultations.
- Include a summary table with detailed results of the intervention, covering the first and second consultations and communication techniques.
- Missing data – how many patients received written and online information? Include descriptive statistics on distributed written materials. This provides essential context for evaluating communication practices.
- Line 196: “Two pharmacy technicians stated that telephone consultations were more effective than those at the counter. They experienced that there was more interaction due to increased privacy and less pressure” – require empirical support? How was “effectiveness” operationalised and measured? Was this based on self-perception, patient feedback, or another metric?
- Lines 205-207: Not clear enough.
- Line 208: How did you measure that? What metric or observation confirms this? Interviewee frequency, verbatim content, or researcher interpretation?
- Line 336: Provide more explanations. Did technicians receive any prior education or standard briefing before the study started? Sampling Bias: The manuscript should acknowledge whether participants were already sensitised to DIM issues (e.g., prior training).
- Counselling Standards: Is there a national or institutional protocol for second dispense counselling? If yes, was adherence measured?
- The manuscript emphasises pharmacy technicians but lacks reflection on how pharmacist involvement could influence outcomes. Reflect on whether pharmacist-led consultations could mitigate some identified barriers (e.g., dealing with complex drug interactions or legal advice about driving).
- Add Implications for Education: The manuscript suggests refresher courses are needed, but doesn't specify the content or format of ideal training programs. Could the authors describe an outline for a continuing education module?
- More concrete policy proposals in conclusion would strengthen the manuscript—for instance, embedding driving fitness alerts into dispensing software, or mandatory follow-up consultations for specific DIMs.
Author Response
Please see the attachment uploaded for reviewer #4.

Reviewer 5 Report
Comments and Suggestions for Authors
The manuscript focuses on the dispensing of driving-impairing medicines in pharmacies. Overall, the structure of the manuscript is appropriate, and it addresses an important topic that is likely to attract readers' interest. However, before publication, some improvements are necessary:
- It would be helpful to include a table listing various types of driving-impairing active substances, not only benzodiazepines and opioids.
- The manuscript could also be improved with the addition of figures—for example, a flowchart illustrating the study process.
Author Response
Please see the attachment uploaded for reviewer #5.

Reviewer 6 Report
Comments and Suggestions for Authors
- References should be placed within parentheses, with the period following the citation. The entire manuscript should be revised accordingly.
- The manuscript should be checked thoroughly for double spacing throughout the text.
- The ellipsis in line 83 needs clarification — its purpose and meaning in this context are unclear.
- Line 102 – it is unclear what is meant by “nivel” and “health base”; readers from other countries may not understand these terms, so further explanation or clarification is recommended.
- This study did not seek ethical approval, and as such, I consider it unsuitable for publication. The authors state that participants provided informed consent; however, informed consent is an official document that must also be approved by an ethics committee. Recommend read: https://journals.lww.com/academicmedicine/fulltext/2014/09000/Standards_for_Reporting_Qualitative_Research__A.21.aspx
Author Response
Please see the attachment uploaded for reviewer #6.

Round 2
Reviewer 2 Report
Comments and Suggestions for Authors
I appreciate the authors' effort to increase the quality of the manuscript.
Reviewer 3 Report
Comments and Suggestions for Authors
The authors have improved the manuscript accordingly and responded to the comments very professionally. The manuscript now qualifies for publication.
regard
Reviewer 4 Report
Comments and Suggestions for Authors
Dear authors,
Thank you for accepting suggestions. The manuscript can be published.
Reviewer 6 Report
Comments and Suggestions for Authors
If editor finds suitable study without ethical approval, which authors explained in depth and I am grateful for this explanation, I find manuscript acceptable for publication